Design of intelligent financial data management system based on higher-order hybrid clustering algorithm

Huang Ling 1
Lu Haitao lu001710@163.com 2
1 School of Management, Wuhan Technology And Business University , Wuhan , China
2 Department of Accounting, Henan Institute of Economics and Trade , Zhengzhou , China
Asif Muhammad
Electronic publication date: 2024 Jan 24
Publication date: 2024
Volume: 10
Electronic Location ID: e1799
Received 2023 Oct 25; Accepted 2023 Dec 15
Copyright: ©2024 Huang and Lu
Copyright year: 2024
Copyright holder: Huang and Lu
License: This is an open access article distributed under the terms of the Creative Commons Attribution License, which permits unrestricted use, distribution, reproduction and adaptation in any medium and for any purpose provided that it is properly attributed. For attribution, the original author(s), title, publication source (PeerJ Computer Science) and either DOI or URL of the article must be cited.
License URL: https://creativecommons.org/licenses/by/4.0/

Keywords: CNN, VAE, Attention mechanism, Deep clustering, Finance risk prediction

Funding: The author received no funding for this work.

==============================
Amid the ever-expanding landscape of financial data, the importance of predicting potential risks through artificial intelligence methodologies has steadily risen. To achieve prudent financial data management, this manuscript delves into the domain of intelligent financial risk forecasting within the scope of system design. It presents a data model based on the variational encoder (VAE) enhanced with an attention mechanism meticulously tailored for forecasting a company’s financial peril. The framework called the ATT-VAE embarks on its journey by encoding and enhancing multidimensional data through VAE. It then employs the attention mechanism to enrich the outputs of the VAE network, thereby demonstrating the apex of the model’s clustering capabilities. In the experimentation, we implemented the model to a battery of training tests using diverse public datasets with multimodal features like AWA and CUB and verified with the local finance dataset. The results conspicuously highlight the model’s commendable performance in comparison to publicly available datasets, surpassing numerous deep clustering networks at this juncture. In the realm of financial data, the ATT-VAE model, as presented within this treatise, achieves a clustering accuracy index exceeding 0.7, a feat demonstrably superior to its counterparts in the realm of deep clustering networks. The method outlined herein provides algorithmic foundations and serves as a pivotal reference for the prospective domain of intelligent financial data governance and scrutiny.

Introduction

With the continuous development of the global economy, refined quantitative analysis of finance has become an indispensable part of decision-making for enterprises and investors. Financial analysis can help investors and clients evaluate the financial health of a company, thereby completing corresponding tasks from a more scientific perspective. Traditional financial analysis methods include financial ratios, financial statement analysis, trend analysis, and comparative analysis. In the study of financial information, the focus needs to monitor the financial risks of companies. Once financial risks occur, especially for listed companies, it will bring huge losses or even bankruptcy to the enterprise. Financial risks may also lead to deterioration of the company’s operating conditions, decline in stock prices, loss of investor confidence, and other issues, thereby affecting the long-term development and interests of the company. Therefore, monitoring the financial condition of a company and evaluating its financial health through more intelligent means has become a research focus in the current financial and financial data management fields (Goodell, Kumar & Lim, 2021).

The types of financial data are complex, especially for large listed companies, which have a massive amount of financial data, such as the company’s profits, cash flow, liabilities, and revenue on the same day. In addition to these numerical types of data, it usually includes text and chart forms of data. Therefore, leveraging the data processing advantages of machine learning and deep learning methods to achieve multimodal data fusion is very important for the intelligent management of financial data. The task of financial analysis is to identify financial risks through the clustering method of multimodal data, which is a clustering problem. Traditional machine learning methods typically rely on manually designed features and have limited ability to handle massive amounts of data. Some traditional clustering algorithms, such as K-means clustering, are very sensitive to the selection of initial cluster centers. Different initial values may lead to different clustering results, and it is necessary to run the algorithm multiple times to obtain stable results. If financial analysis involves textual data, traditional machine learning methods may not be as flexible in processing and analyzing textual features as deep learning methods.

Deep learning models are usually more suitable for situations with complex data modalities and diverse properties, such as auto-encoder. Autoencoder is an unsupervised learning model commonly used for feature learning and data dimensionality reduction (Merah, Aliouat & Harbi, 2023). Deep models such as convolutional neural network (CNN), LSTM, and other modeling methods that excel in processing time series data (Shi, Wang & Zhao, 2022) can capture implicit features in time series, thereby completing tasks such as data classification, regression, and prediction. These methods are used in fields such as stock prediction and futures analysis in the financial field. The design of an intelligent financial system data system, in addition to ensuring basic data storage and visualization functions, also requires a certain level of intelligent decision-making ability. Using deep learning methods to identify financial risks based on multimodal data is crucial for the construction of financial systems. This article proposes a clustering model based on deep networks for intelligent management of financial data, aiming to achieve intelligent clustering of financial data risks. The specific contributions are as follows.

(1) Synergizing multiple techniques: Our method amalgamates the variational autoencoder (VAE) and the attention mechanism, facilitating multidimensional data clustering and enhancing financial data analysis. The combination of these techniques maximizes the performance of each method and improves data clustering capabilities.

(2) Holistic financial data analysis: Our approach not only enables high-quality clustering analysis of financial data but also supports intelligent financial system management, substantially reducing human intervention for sustainable development.

(3) Model construction and training: By summarizing the characteristics of current clustering and class analysis models, an ATT-VAE clustering model based on VAE and attention mechanism was established. The model was trained and tested on multiple datasets, such as AWA. The experimental results showed that our ATT-VAE method has better performance and can better analyze financial and financial risk data.

The subsequent sections of this article are organized as follows: ‘Related works’ presents related work, while ‘Methodology’ introduces the proposed methods, VAE, and the attention mechanism. Section ‘Experiment Setup and Result Analysis’ provides details about the experiments, and ‘Discussion and Conclusion’ offers conclusions.

Related Works

At present, many achievements have been made in financial management and clustering research, which mainly focus on the quantification of financial data indicators and the improvement of clustering performance. Quantifying key indicators in the intelligent management of financial data and providing quantitative references for decision-makers are the current research focuses. For the research on clustering methods, the main focus is on improving and enhancing traditional clustering methods. With the development of deep learning technology, clustering based on deep learning methods has also become a research hotspot. Therefore, this section will introduce financial data management, traditional clustering methods, and deep learning clustering methods from three aspects.

Intelligent management of financial data

The realm of financial risk early warning holds a pivotal standing within the purview of financial management and investment decision-making. As the sands of time have drifted by, the focus of inquiry and the arsenal of analytical tools have undergone a progressive evolution. It has transitioned from the examination of a scant number of company samples in its nascent stages to the comprehensive scrutiny of a multitude of companies replete with complete datasets. Concurrently, the means of analysis has advanced from the rudimentary utilization of financial indicator ratios to the construction of secondary indicators. It has further embraced the infusion of mathematical and statistical methodologies to give rise to multivariate discriminant analysis models epitomized by the Z models. Presently, the landscape resonates with the widespread adoption of diverse machine learning and deep learning models (Lin, Hu & Tsai, 2011). Ohlson’s seminal work, expounded in Ohlson (1980), unveiled a logistic-based early warning model. This model, distilled through the meticulous examination of over 2,000 solvent and insolvent companies, aspires to forecast the probability distribution of corporate bankruptcy. Shin, Lee & Kim (2005) embarked on a pioneering exploration into the application of support vector machine models within the sphere of machine learning for financial risk prognosis. Empirical evidence underscored the supremacy of the SVM model over traditional multivariate discriminant analysis and logit models in the realm of financial risk assessment. With the burgeoning computational prowess, the mantle was passed to neural network models and CNN models, both of which garnered resounding success within this domain (Atiya, 2001). Odom Sharda (1990) championed a financial risk early warning model leveraging artificial neural networks. They painstakingly assembled a dataset teeming with an equitable number of precarious and stable companies for in-depth analysis, ultimately yielding commendable predictive outcomes. Marcano-Cedeno et al. (2011) ventured into the realm of metaplasticity neural networks (AMMLP) and engineered an enhanced ANN model for credit default risk evaluation. Furthermore, Hosaka (2019) introduced a novel approach, transmuting the financial indicators of insolvent companies into grayscale images. Subsequently, they harnessed a CNN deep learning model to predict corporate bankruptcy risks with promising outcomes.

Research on traditional clustering methods

Traditional clustering algorithms, often referred to as early clustering algorithms, exhibit commendable performance on small-scale, low-dimensional datasets. Their development has reached a mature stage, owing to their intuitively comprehensible principles and straightforward implementation. These methods find extensive utility across various domains, particularly in the realm of image processing. Prominent among these traditional clustering algorithms are the K-means algorithm and the spectral clustering algorithm, among others. Typically, these algorithms take as input a data matrix composed of image or text features, employing diverse clustering strategies to gauge the similarity relationships among these features and subsequently generating clustering outcomes (Ahmed, Seraj & Islam, 2020). The K-means clustering algorithm halts when it attains a local optimum solution. Notably, it is tailored for numerical data clustering. It boasts the virtues of simplicity and efficiency, characterized by low algorithmic complexity. However, it bears certain drawbacks, such as sensitivity to predetermined values for the number of clusters, vulnerability to noise and outliers, and suboptimal performance on datasets with non-spherical clusters. In an endeavor to enhance the K-means algorithm’s performance, the DIANA split hierarchical clustering algorithm was introduced, treating the provided data as a cluster structure and progressively partitioning the most recently formed cluster into smaller clusters based on cluster diameter or average dissimilarity (Patnaik, Bhuyan & Rao, 2016).

Guha, Rastogi & Shim (1998) introduced the CURE algorithm, an improved hierarchical clustering method that leverages a representative subset of points to depict a cluster, departing from the conventional approach of using all points or a single center of mass. This modification renders it more resilient to isolated points and equips it to identify clusters characterized by complex shapes and varying sizes. The incorporation of fuzzy set theory into hard clustering algorithms, assigning each sample a certain probability of belonging to a particular class, has birthed fuzzy clustering algorithms (Nayak, Naik & Behera, 2014). Krinidis & Chatzis (2010) contributed a robust C-mean clustering algorithm for fuzzy local information, introducing a fuzzy local neighborhood factor to amalgamate local spatial and grayscale information, thereby diminishing the clustering method’s sensitivity to noise.

Seal, Karlekar & Krejcar (2020) introduced a fuzzy clustering technique employing nonlinear distances, substituting s-distance for the Euclidean distance metric, resulting in more robust natural clustering outcomes. Beyond K-means approaches, spectral clustering and its derivatives have garnered significant traction in contemporary clustering methodologies. Wang, Qian & Davidson (2014) introduced constrained spectral clustering, augmenting spectral clustering with additional sub-information to bolster clustering results. By leveraging pairwise constraints, this approach tackles challenging segmentation tasks by determining whether two points are linked based on the introduced edge information. Zhou, Lin & Wang (2023) proposed the statistical correlation coefficient of a single-valued neutrophil set and explained the relevant mathematical principle; Xu et al. (2023b) proposed the traffic data input algorithm to provide the relevant algorithm support; Chen et al. (2023) proposed the target detection based on consistency and dependence. Chen, Song & Bai (2010) proposed a parallel spectral clustering approach for deployment in distributed systems. They juxtaposed two methods for approximating dense matrices to address concerns related to memory consumption and computational time scalability in spectral clustering. Ultimately, they opted to retain the nearest neighbors to sparsify the matrix, a strategy applicable to solving problems within distributed systems. Traditional clustering methodologies can also harness representation learning techniques for feature extraction, including subspace representation learning and deep network representation learning. This circumvents the limitations of conventional methods when confronted with high-dimensional data.

Research on clustering algorithms based on deep learning

Deep learning-based clustering methods are categorized into generative model-based methods and discriminative model-based methods based on the nature of the network model and the results of these two types of methods can be subdivided again as shown in Fig. 1.

Deep clustering methods encompass a diverse array of techniques, including those predicated on VAE, GAN, intricate deep models, and GNN. The architecture of autoencoder-based clustering methods typically comprises two fundamental components: the autoencoder module and the similarity measure module (Song, Liu & Huang, 2013). Various methods adopt distinct training strategies, with one of the pioneering approaches being the inception of deep embedding clustering (Xie, Girshick & Farhadi, 2016). This method disentangles the dimensionality reduction process from the similarity metric computation within the framework. It commences with the acquisition of a proficient encoder model via self-encoder training, subsequently proceeding to joint training of the encoder and similarity metric modules. However, this approach renders the embedded features overly reliant on the initialized encoder model, which can exert an impact on clustering outcomes. Building upon this foundation, a fusion between the traditional K-means method and deep clustering methodology was realized, culminating in the co-optimization of dimensionality reduction and similarity metrics, thereby yielding more optimized results (Yang, Fu & Sidiropoulos, 2017). Presently, deep learning-based clustering methodologies predominantly hinge on similarity metrics. In pursuit of neighborhood relationships, these methods typically employ local constraints during the similarity metric computation process. While local constraints effectively ascertain the similarity of points within clusters, they may falter in precisely distinguishing the class attributes of points positioned at the cluster periphery.

Consequently, this can result in indistinct cluster boundaries within the feature space (Min, Guo & Liu, 2018). Moreover, approaches grounded in self-coder models can also be amalgamated with spectral clustering, subspace clustering, and other techniques. The choice of neural networks for encoding can significantly impact outcomes, with convolutional neural networks often outperforming fully connected neural networks (Santhosh, Dogra & Roy, 2021; Xu et al., 2023a).

The evolution of segment clustering research clearly demonstrates that contemporary deep learning methods offer superior practical utility when juxtaposed with traditional clustering approaches. Furthermore, the amalgamation of VAE with CNN and other techniques augments the robustness of self-supervised and semi-supervised models. Given the sheer volume of financial data, along with the challenges of missing data and the impracticality of manual labeling, the adoption of advanced deep clustering methods assumes paramount significance in the domain of financial system management. Consequently, this manuscript proffers a novel proposition: the enhancement of unsupervised analysis of financial data through the synergistic integration of VAE and existing CNN methodologies.

Methodology

After completing the analysis of data characteristics and the introduction of clustering methods, this article intends to build a network model based on existing deep learning methods to complete the clustering analysis of financial data.

Convolutional neural networks

CNN is an influential deep learning model extensively deployed in computer vision tasks, encompassing image classification, target detection, and image segmentation, the watershed moment for CNNs occurred in 2012 with the advent of the AlexNet network (Krizhevsky, Sutskever & Hinton, 2012), which solidified the standing of convolutional neural networks in the domain of deep learning. In this study, we opt to employ AlexNet for data analysis. In addition to the convolution operation outlined in Eq. (1), AlexNet augments network generalization performance by incorporating local normalization (LRN). The LRN is computed as depicted in Eq. (2): (1) I∗Kx,y= ∑i=−aa ∑j=−bbIx+i,y+j⋅Ki,j

(2) Rx,y,k=Ax,y,kκ+α∑i=max0,k−n2minN−1,k+n2Ax,y,i2β

In Eq. (1), I represents the input data, K is the convolution kernel, i, and j are the coordinates of the convolution kernel, and a and b are the radii of the convolution kernel. For AlexNet’s special link LRN, R in Eq. (2) represents the output response, A is the original unnormalized response, and the rest of the adjustable parameters are used to control the degree of normalization and parameters, such as the window and the number of channels. In addition to this, models such as ResNet, GoogleNet, etc., are widely used methods in CNN-like networks (Targ, Almeida & Lyman, 2016). The clustering ability of the model can be greatly enhanced by augmenting the current information through convolutional neural networks.

Self-Encoder models and Variational Self-Encoders

Self-Encoder (AE) models and Variational Self-Encoders (VAE) both belong to the realm of unsupervised learning models that find prominent utility in deep learning for uncovering latent data representations. They serve a multitude of purposes, including feature extraction, data compression, and dimensionality reduction. AE constitutes a neural network architecture comprising two principal components: an encoder and a decoder. The encoder serves to map input data into a lower-dimensional representation while the decoder endeavors to reconstruct this lower-dimensional representation back into the original input data. The primary objective of AE is twofold: to achieve accurate reconstruction of the input data and to distill essential features of the input data within the low-dimensional representation generated by the encoder. This is illustrated in Fig. 2.

Figure 1 The deep clustering methods.

Figure 2 Structure of self-encoder.

The encoding and decoding of the encoder is done through function mapping; after completion, the model training needs to be realized through the definition of the loss function and objective function, which are defined as shown in Eqs. (3) and (4): (3) Lx,x′=1n∑i=1nxi−xi′2

Where: Lx,x′ denotes the loss function, representing the original input x and reconstructed input x′ mean square error between the original and reconstructed inputs.

The goal of this loss function is to minimize the difference between the reconstructed data and the original input, allowing the AE to learn a valid representation of the data. However, depending on the specific task and data type, other loss functions can be chosen, such as the cross-hashing loss. The objective function is then expressed by Eq. (4). (4) Θ∗= argminΘ1N∑i=1NLxi,xi

where: Θ∗ denotes the model parameters to be optimized to minimize the loss function. N denotes the number of samples in the training dataset. Lxi,x′i denotes the loss function that measures the number of samples in the first i original input of the first training sample xi and reconstructed input x′(i) between the original and reconstructed inputs of the first training sample and the mean square error or other loss measures. The optimal output of the model can be obtained by optimizing the objective function and the loss function. From the above process, it can be seen that VAE utilizes probability graph models combined with encoders and decoders to learn potential representations of data distribution, minimize reconstruction losses and KL divergence during the training process, generate features with continuity and diversity, and improve model performance by integrating these features with the original features.

Overall framework of the attention-based VAE

VAE has the following advantages over ordinary AE: a VAE is a generative model that learns valid data representations and generates new samples; its latent space is continuous and interpretable, allowing operations such as interpolation, sampling, and so on, to generate diverse samples; the latent representations are more easily interpretable, which helps with a variety of downstream tasks; and the generated samples are typically of higher quality because the VAE generates samples by learning the latent distributions, rather than just replicating the training data points. The structure of the VAE is shown in Fig. 3:

Figure 3 The framework for the VAE.

Figure 3 underscores the key distinction between VAE and AE. In VAE, an additional statistical module comes into play, which corresponds to the encoder component of the VAE. At the same time, the generator aligns with the decoder, symmetrically positioned with respect to the encoder. The encoder undertakes the computation of mean and variance for each input, assigning a normal distribution to each input data point. It is essential to ensure that the variance in this normal distribution is not zero, as a zero variance would lead to a loss of randomness, making it challenging for the decoder to reconstruct the samples effectively in the presence of noise. During the sampling process, as sampling itself is a non-differentiable operation, the sampled result is not directly amenable to gradient-based optimization. To circumvent this issue, the re-parameterization technique is employed, allowing for the design of a differentiable sampling operation. This enables the optimization of the mean–variance model in reverse. The probability distribution of the encoder is encapsulated in Eq. (5). (5) qΘz∣x=NμΘx,σΘx2

where Θ denotes the parameters of the encoder, and μΘ(x) and σΘ(x) denote the mean and standard deviation, respectively. Latent space sampling. (6) z=μΘx+σΘx⋅ϵ

where ϵ is the random noise sampled from the standard normal distribution. The normal distribution feature is also introduced in the decoder section with the conditions shown in Eq. (7): (7) pΦx∣z.

This means that given the potential variables z that generates the data in the case of x of the conditional distribution. Φ denotes the parameters of the decoder.

In this study, in order to enhance the model performance, we add the attention mechanism, which is a technique widely used in deep learning to enhance the neural network’s attention to certain parts of the input data, thus improving the model performance. In the attention mechanism, there are usually three key components, query, key, and value features, and the model enhancement is achieved by attention score, attention weight and weighted sum, Q denotes the query and K denotes the key. The attention mechanism’s query determines the focus, keys provide context, and values hold relevant information, collectively enabling the model to dynamically weigh and incorporate different parts of the input sequence for improved context-aware processing. Then the attention score can be expressed as Eqs. (8)–(10): (8) AttentionQ,K=Q⋅Kdk

(9) Attention_WeightsQ,K=softmaxAttentionQ,K

(10) Attention_OutputQ,K,V= ∑iAttention_WeightsQ,Ki⋅Vi

where dk denotes the dimension of the key. The core idea of the attention mechanism is to assign the weights of the values based on the relationship between the query and the keys in order to better capture the relevant information of the input data in different tasks and contexts. This dynamic attention mechanism enables the neural network to handle sequential data better and improves the generalization ability of the model. The model based on the attention mechanism established in this article is shown in Fig. 4:

Dynamic attention mechanisms in Fig. 4 likely improve sequential data processing by allowing the model to adaptively assign varying importance weights to different parts of the input sequence. This enables it to focus on relevant information at different time steps, capture long-range dependencies, and better handle varying contextual patterns. In Fig. 4, after completing the data input, we realized the feature extraction enhancement of the data by VAE and completed the final model optimization by using the attention mechanism on the right side to realize the clustering judgment of the system.

Experiment Setup and Result Analysis

In this section, we will provide a detailed introduction to the characteristics of the dataset used, as well as the experimental process and test results of actual data.

Datasets

Considering the characteristics of the clustering method and the characteristics of the data used, the data used in this article include the following five:

AwA (https://cvml.ist.ac.at/AwA/) has a total of 5,814 instances and consists of three modalities, local self-similarity features, SIFT features and SURF features, and contains 10 clusters. Scene-15 (Fei-Fei & Perona, 2005) has 3,000 instances and consists of three modalities: LBP features, GIST features, and CENTRIST features containing 15 clusters. CUB (http://www.vision.caltech.edu/datasets/cub_200_2011/) contains 50 clusters totaling 2,889 data instances. Two modalities consist of 1,024-dimensional image features extracted by GoogleNet and 1,024-dimensional corresponding text features (Reed, Akata & Lee, 2016). Flowers (http://www.robots.ox.ac.uk/v˜gg/data/flowers/102/) contains 50 clusters totaling 3,235 data instances. The two modalities consist of GoogleNet-extracted 1,024-dimensional image features and 1,024-dimensional corresponding text features. Both image features are removed at the time of input. The specific information of the adopted dataset is shown in Table 1.

Figure 4 Framework of the proposed ATT-VAE.

Experiment details

After completing the data collection of the dataset, it is necessary to determine the relevant details of the study, mainly including model evaluation indexes, model training process and so on. In order to evaluate the experimental results, we adopt three evaluation indexes: Accuracy (ACC), Normalized Mutual Information (NMI) and Adjusted Rand Index (ARI), all of which are higher to indicate better performance. Clustering accuracy is used to compare the clustering assignment labels with the true labels provided by the data. (11) ACC=∑i=1Nδsi,mapciN

where si denotes the true label of the first i, the first sample, and ci denotes the label assigned by the clustering algorithm to the i the label assigned to the first sample by the clustering algorithm, and N is the total number of data, and mapci Calculate ci to si The mapping between δ is determined by the following formula: (12) δx,y=1ifx=y0otherwise

NMI is defined as the following equation. (13) NMI=MIS,CmaxHS,HC

where S, C are two different labels of the same sample, i.e., the true label and the cluster assignment label, and MIC,C′ The NMI results do not change depending on the arrangement of the clusters. They are normalized to the cluster assignment labels. H(⋅) The results of NMI do not change depending on the arrangement of the clusters. They are normalized to the range of 0 for uncorrelated and 1 for perfectly correlated [0, 1]. The results of NMI do not change according to the arrangement of clusters, and they are normalized to the range of 0 for no correlation and 1 for perfect correlation.

Table 1 The specification of the dataset.

Dataset	Modal	Samples	Cluster	
AwA	3	5,814	10	
Scene	3	3,000	3	
CUB	2	2,889	2	
Flower	2	3,235	2	

RI (Rand Index) represents the rate of correct decision-making and is defined as (14) RI=TP+TNTP+TN+FP+FN

where TP is the true positive,TN is the true negative, FP is the false positive and FN is the false negative. The Rand Index has values between [0, 1]. The Rand Index (RI) has a value between 1 and 2, and the RI is 1 when the two classifications match.

Unsupervised deep embedding clustering (DEC) (Xie, Girshick & Farhadi, 2016), improved DEC (IDEC) (Guo, Gao & Liu, 2017) and deep neural networks for spectral clustering (SpectralNet) (Shaham, Stanton & Li, 2018), deep canonically correlated auto-encoders (DCCAE) (Wang, Arora & Livescu, 2015) used for clustering analysis in clustering research are the more widely used methods that are more mature and represent their respective fields. Methods, so this study chooses the above methods for comparison. After confirming the dataset and related indexes, we trained the model and the training methods used for different models are similar, and the specific steps are shown in Algorithm 1 :

Algorithm 1: Training process of ATT-VAE for clustering Input: AWA dataset, Scene dataset, CUB dataset, Flower dataset	
Initialization.	
Define the ATT-VAE.	
Define the hyperparameters and Initialization.	
Define the loss function.	
Define the optimizer: Adam optimizer.	
Define the number of training epochs.	
Feature extraction.	
Using the original features in the dataset	
Pre-training: Initialize the pre-training step counter.	
while pre-training step counter <pre-training steps do	
Sample a batch of data.	
Feed data to the ATT-VAE framework.	
Update model.	
Counter++;	
End	
Parameters tuning	
Tuning counter definition TT.	
while TT<Preset iteration do	
Feed sample data to the proposed network.	
Loss and gradients calculation.	
Model updated.	
Compute ACC, NMI and RI	
Save the optimal model	
end	
Output: Trained ATT-VAE	

After completing the model building and training of the relevant data, we performed statistics for the model.

Experiment result and analysis

Based on the pertinent metrics and model training procedures elucidated in ‘Datasets’ and ‘Experiment details’, we subjected the data to diverse test sets. We shall now elucidate the detailed clustering outcomes for each dataset. Table 2 and Fig. 5 encapsulate the clustering results for the AWA dataset. Notably, the introduction of the deep network has manifestly enhanced clustering performance, with the proposed method showcased in this study surpassing conventional approaches in the present stage across all three metric categories: ACC, NMI, and RI. Indeed, the proposed method outperforms common techniques across all three metric types at this juncture.

Table 2 The comparison result of three indicators concerning AWA datasets.

	DEC	IDEC	SpectralNet	DCCAE	Proposed	
ACC	0.21	0.23	0.21	0.25	0.26	
NMI	0.04	0.07	0.04	0.1	0.11	
RI	0.14	0.16	0.15	0.17	0.19	

After completing the analysis of the AWA data, we similarly analyzed the data on the three datasets SCENE, CUB, and Flower, the results of which are shown in Figs. 6, 7 and 8, and the corresponding data results are given accordingly in Tables 3, 4 and 5.

It is not difficult to find from the comparison of the data in the figure that the ATT-VAE method proposed in this article performs well on different datasets. Among them, the advantage is not obvious in the SCENE dataset, while it shows obvious advantages in the latter two types of modal data with less data. This further indicates that the ATT-VAE method has certain performance advantages and can complete clustering in small samples and low dimensional modal data.

After completing the comparison of multiple methods, this article also carries out batch size comparison experiments of the proposed method under different datasets, which are tested through eight batch sizes ranging from 2 to 128, and the corresponding boxplots obtained are shown in Fig. 9:

Figure 5 The comparison result of three indicators concerning AWA datasets.

Figure 6 The comparison result of three indicators concerning SCECE datasets.

Figure 7 The comparison result of three indicators concerning CUB datasets.

Figure 8 The comparison result of three indicators concerning Flower datasets.

In Fig. 9, it is evident that the variance in clustering results across different batch sizes is relatively modest. This observation underscores the inherent robustness of the method advanced in this article.

Examining the data presented in the icon, it becomes evident that the ATT-VAE model, as introduced in this article, boasts commendable generalization prowess and data clustering acumen. This is particularly conspicuous in the case of the CUB and Flower datasets, which feature fewer attributes and categories. In such scenarios, the clustering efficacy is notably pronounced. This attribute bodes well for the application of this method in the financial analysis of low-dimensional data characteristic of financial system analysis. As an extension of this approach, the article now extends its ambit to the realm of economic and financial research. Leveraging solvency, operational capacity, and profitability indicators as provided by this source, the model undertakes risk analysis through clustering, stratifying entities into high-risk and low-risk categories. This dataset is denoted as the Finance database. The results pertaining to ACC and NMI under this database are depicted in Fig. 10.

Table 3 The comparison result of three indicators concerning AWA datasets.

	DEC	IDEC	SpectralNet	DCCAE	Proposed	
ACC	0.17	0.25	0.46	0.35	0.52	
NMI	0.17	0.19	0.45	0.38	0.49	
RI	0.14	0.18	0.23	0.27	0.35	

Table 4 The comparison result of three indicators concerning CUB datasets.

	DEC	IDEC	SpectralNet	DCCAE	Proposed	
ACC	0.21	0.31	0.19	0.17	0.48	
NMI	0.34	0.41	0.38	0.33	0.58	
RI	0.18	0.23	0.15	0.16	0.46	

Table 5 The comparison result of three indicators concerning Flower datasets.

	DEC	IDEC	SpectralNet	DCCAE	Proposed	
ACC	0.18	0.28	0.27	0.21	0.53	
NMI	0.35	0.43	0.42	0.41	0.69	
RI	0.15	0.26	0.19	0.18	0.49	

Figure 9 The ACC for the different datasets using different batch sizes.

Figure 10 The ACC and NMI result with the Finance datasets.

In Fig. 10, we can see that as the complexity of the deep network gradually increases, i.e., the number of modules used in the model increases, the performance of IDEC improves compared to DEC. And through the self-established model in this article, with the addition of the attention mechanism, the clustering accuracy of the model is also increasing, and the accuracy of the proposed method in this article has certain advantages.

Discussion and Conclusion

This study introduces a deep clustering model based on ATT-VAE, aiming to achieve adaptive clustering of financial risks in the field of financial data. We conduct performance testing on general high-dimensional low-feature clustering data and conduct actual demand testing based on actual financial risk data. The experimental results show that compared to current clustering methods such as DEC and IDEC, this method performs well in clustering multimodal data. This method effectively enhances the output of the VAE model by introducing an attention mechanism, achieving more precise clustering. In the test of real financial data, this method shows its progressiveness, reaching a clustering accuracy of more than 70%, providing strong support for future financial analysis and data management. Anticipating forthcoming work, we envision broadening the adaptability of the model to embrace a plethora of financial data formats while refining its data processing capabilities.

Furthermore, we intend to delve into advanced optimizations for the clustering network and the augmentation of model capabilities through the incorporation of techniques such as reinforcement learning. Although the VAE model employed in this investigation exhibits robust clustering capabilities, we will explore avenues to render the model more streamlined in the future. Furthermore, for financial data characterized by judiciously selected and processed features, the data analysis may potentially be carried out by a more lightweight clustering model. Thus, the extraction of features from the model proposed in this article, as well as from related deep clustering models, constitutes a central focus for future research, with the objective of attaining more comprehensive features.

Supplemental Information

Supplemental Information 1 This is a code

Additional Information and Declarations

Competing Interests

Author Contributions

Data Availability

The author declares that he has no competing interests.

Ling Huang conceived and designed the experiments, performed the experiments, analyzed the data, performed the computation work, prepared figures and/or tables, and approved the final draft.

Haitao Lu conceived and designed the experiments, performed the experiments, analyzed the data, performed the computation work, prepared figures and/or tables, authored or reviewed drafts of the article, and approved the final draft.

The following information was supplied regarding data availability:

The code is available in the Supplementary File.

The datasets are available at Zenodo:

- Humbert, J., Williams, K., & Onthank, K. (2022). Underwater Camera Trap Photo Data (v1.0) [Data set]. Zenodo. https://doi.org/10.5281/zenodo.6410362.

- SCENE. (2021). SCENE open data pilot [Data set]. Zenodo. https://doi.org/10.5281/zenodo.4562601.

- Wang, X. (2020). CUB-200 Speech captions (Part-I) [Data set]. Zenodo. https://doi.org/10.5281/zenodo.3972388.

- Sebastian Hönel. (2023). Metrics As Scores Dataset: The Iris Flower Data Set (v1.1) [Data set]. Zenodo. https://doi.org/10.5281/zenodo.7669664.

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
