# Peer review of "Design of intelligent financial data management system based on higher-order hybrid clustering algorithm"

_PeerJ Computer Science, doi:10.7717/peerj-cs.1799_

## Round 0.1 · original submission · Major Revisions

Dear Author,

Your paper has been reviewed with interest by experts in the field and you will see that they have a couple of comments to be considered for the improvement of your paper. Therefore, Please carefully revise the paper and resubmit; please also improve the language of the article.

thank you

**Language Note:** The Academic Editor has identified that the English language must be improved. PeerJ can provide language editing services - please contact us at copyediting@peerj.com for pricing (be sure to provide your manuscript number and title). Alternatively, you should make your own arrangements to improve the language quality and provide details in your response letter. – PeerJ Staff

Reviewer 1 ·

Basic reporting

(1) The abstract of this paper lacks specific experimental data, which the author should add;
(2) There are too few references and introductions to traditional methods in the introduction section;
(3) The author may add a paragraph before each section to describe the general content of that section;

Experimental design

(4) What role do AE and VAE play in the model and what do they contribute to this article?
(5) The various models and data parameters can be tabulated in section 4.1;
(6) What is the role of the three key components of the attention mechanism in the model?

Validity of the findings

(7) The three measurement categories in section 4.3 need to be explained;
(8) Figure 10 shows how the accuracy of the model changes with increasing complexity. Please explain the reasons for the changes;
(9) I would suggest that the author add some excellent journal articles from recent years.

Additional comments

In the context of expanding financial data, the importance of predicting potential risks through AI methods has steadily increased. In order to realize robust financial data management, this paper studies the field of intelligent financial risk prediction deeply within the scope of system design. In the experiment, the authors ran a set of training tests on the model using different data sets. Compared to publicly available datasets, the results significantly highlight the model's commendable performance, outperforming numerous deep clustering networks at this critical time. However, the author still needs to modify the points mentioned to improve.

·

Basic reporting

In this study, the authors has introduce an innovative deep clustering paradigm using ATT-VAE to promote adaptive clustering of financial risks in the field of financial data. The experimental results presented in this paper are good to validated the method described in this manuscript. By utilizing different clustering methods and enhancing the VAE model by integrating attention mechanisms, the method further proves its power in real financial data evaluation, with clustering accuracy of more than 70%. However, there are still the following shortcomings in the article:
1. The author may propose a specific method model in the topic;
2. The author contribution is introduced in the section where the third point of model validation requires rewriting a schema to highlight the contribution;
3. The related work section has been too long. I would suggest that the author revise and simplify the section to highlight the main points;
4. How does the author process the data to achieve unsupervised analysis?
5. How does the dynamic attention mechanism in Figure 4 better process sequential data?
6. VAE can learn the underlying distribution, can it embody the core innovation?
7. The description of Figure 7, 8, and 9 in the experimental part of the paper is too little, and the advantage of the experimental results is not highlighted;
8. The discussion and conclusions in Part 5 seem to have much in common with the abstract and need to be revised;
9. The writer should polish and strengthen the language as a whole.

Experimental design

.

Validity of the findings

.

---

## Round 0.2 · accepted · Accept

Based on the peer review from the experts in the second round, I'm pleased to inform you that they are satisfied with your revised version and therefore we are recommending it for publication. Congratulations

Reviewer 1 ·

Basic reporting

All changes have been completed.

Experimental design

All changes have been completed.

Validity of the findings

All changes have been completed.

·

Basic reporting

This paper has been well revised.

Experimental design

This paper has been well revised.

Validity of the findings

This paper has been well revised.

Additional comments

This paper has been well revised.